# MicroRNA Sequencing Revealed *Citrus* Adaptation to Long-Term Boron Toxicity through Modulation of Root Development by miR319 and miR171

**DOI:** 10.3390/ijms20061422

**Published:** 2019-03-21

**Authors:** Jing-Hao Huang, Xiong-Jie Lin, Ling-Yuan Zhang, Xian-Da Wang, Guo-Cheng Fan, Li-Song Chen

**Affiliations:** 1Institute of Plant Nutritional Physiology and Molecular Biology, College of Resources and Environment, Fujian Agriculture and Forestry University, Fuzhou 350002, China; jhuang1982@126.com; 2Pomological Institute, Fujian Academy of Agricultural Sciences, Fuzhou 350013, China; linxj_019@163.com (X.-J.L.); 06sansha@163.com (X.-D.W.); guochengfan@126.com (G.-C.F.); 3Fujian University of Traditional Chinese Medicine, Fuzhou 350122, China; daph2220@126.com; 4Fujian Provincial Key Laboratory of Soil Environmental Health and Regulation, College of Resources and Environment, Fujian Agriculture and Forestry University, Fuzhou 350002, China; 5The Higher Educational Key Laboratory of Fujian Province for Soil Ecosystem Health and Regulation, College of Resources and Environment, Fujian Agriculture and Forestry University, Fuzhou 350002, China

**Keywords:** *Citrus*, microRNAs, boron toxicity, lateral root formation, stem cell maintenance

## Abstract

Boron (B) toxicity in *Citrus* is a common physiological disorder leading to reductions in both productivity and quality. Studies on how *Citrus* roots evade B toxicity may provide new insight into plant tolerance to B toxicity. Here, using Illumina sequencing, differentially expressed microRNAs (miRNAs) were identified in B toxicity-treated *Citrus sinensis* (tolerant) and *C. grandis* (intolerant) roots. The results showed that 37 miRNAs in *C. grandis* and 11 miRNAs in *C. sinensis* were differentially expressed when exposed to B toxicity. Among them, miR319, miR171, and miR396g-5p were confirmed via 5′-RACE and qRT-PCR to target a myeloblastosis (MYB) transcription factor gene, a SCARECROW-like protein gene, and a cation transporting ATPase gene, respectively. Maintenance of SCARECROW expression in B treated *Citrus* roots might fulfill stem cell maintenance, quiescent center, and endodermis specification, thus allowing regular root elongation under B-toxic stress. Down-regulation of MYB due to up-regulation of miR319 in B toxicity-treated *C. grandis* roots might decrease the number of root tips, thereby dramatically changing root system architecture. Our findings suggested that miR319 and miR171 play a pivotal role in *Citrus* adaptation to long-term B toxicity by targeting *MYB* and *SCARECROW*, respectively, both of which are responsible for root growth and development.

## 1. Introduction

Boron (B) is one of the important micronutrients that play vital roles in plant growth, development, and reproduction [1,2]. Although, its exact role in higher plants remains to be uncovered, B is involved in diverse biochemical and physiological processes, such as primary cell wall synthesis and secondary lignification, membrane integrity maintenance, carbohydrate and nucleic acid metabolisms, signaling, and gene expression [3]. 

Given the narrow concentration range suitable for plant growth, B would easily produce toxicity to plants when soils contain excessive B [4]. B toxicity, in agricultural production, occurs generally in B-rich soils or in soils where B accumulates due to continued inflow of industrial pollutants, directly affecting yield and quality of various horticultural plants around the world [5].

Evergreen subtropical *Citrus* plants are sensitive to B excess [6]. Inappropriate application of B fertilizer in *Citrus* orchards, especially under low rainfall conditions, can result in over accumulation of B in the older mature leaves, generating typical B-toxic effects. In fact, B toxicity in *Citrus* has recently become a common physiological disorder leading to reductions in both productivity and tree vigor [6,7,8]. A recent investigation by Li et al. [9] showed that 74.8% of 319 leaf samples from ‘Guanximiyou’ pummelo (*C. grandis*) orchards located in Pinghe county of southern Fujian, China, contained excess B due to the abuse of B fertilizer.

In many plant species, a primary phenotypic effect of B toxicity is root growth inhibition, although the precise mechanism for root growth disruption by B is still poorly understood [3]. One hypothesis for barley in the early 21st century is that B might interfere with transcription and/or translation of mitosis-related proteins by binding to cis hydroxyls on ribose molecules, thereby reducing root expansion [10]. Later, the molecular basis of root growth inhibition in *Arabidopsis* was reported by Aquea et al. [11]. They found that B toxicity modulates the expression of genes involved in ABA signaling and water transport, triggering a water-stress response in the roots.

*Citrus* tolerance to B-toxicity varies with cultivars and/or species [8,12]. However, the mechanisms underlying tolerance to B-toxicity in woody *Citrus* seem to differ from those in herbaceous plant species because lowered total B levels have not been observed in the leaves or roots of tolerant species when compared to intolerant ones under B excess conditions [8,13,14]. To date, investigations were mainly focused on the effects of B toxicity on the above plant parts. How *Citrus* roots evade B toxicity by gene regulation is poorly studied.

Derived from non-coding RNA genes, microRNAs (miRNAs) are 21- to 24-nt long small RNAs, which negatively regulate gene expression at the post-transcriptional level by guiding cleavage of target mRNAs [15] or through translation inhibition [16]. Growing evidence demonstrates that plant miRNAs play a pivotal role in the response to B stresses [17,18]. Previously, Huang et al. [19] reported that, in *Citrus* leaves, miR397a is involved in the mitigation of B toxicity to the phloem via modulating secondary cell-wall biosynthesis of the xylem, suggesting miRNA regulation in *Citrus* adaptation to long-term B toxicity. Nonetheless, little information about B-toxicity-responsive miRNAs is available in *Citrus* roots.

With the objective of exploring how *Citrus* roots evade B toxicity, we reported here identification of miRNAs from the roots of B-tolerant (*C. sinensis*) and B-intolerant (*C. grandis*) *Citrus* species [8] treated with different B levels. On this basis, predicted targets of the candidate miRNAs were verified via modified 5′-RACE and qRT-PCR. Our results provide a molecular basis for plant tolerance to B toxicity.

## 2. Results

### 2.1. B-Toxic Treatment on Root Development

The average diameter of *C. sinensis* and *C. grandis* roots under control were 5.58 and 9.00 mm, respectively. B-toxic treatment had no significant effect on average diameter of either *C. sinensis* (6.75 mm; *p* = 0.402) or *C. grandis* roots (8.12 mm; *p* = 0.268). There was no significant difference in either total root length (*p* = 0.075 and 0.894 for *C. grandis* and *C. sinensis*, respectively) or fork number (*p* = 0.46 and 0.914 for *C. grandis* and *C. sinensis*, respectively) between the two B treatments for each *Citrus* species. B-toxic treatment had no influence on root tip numbers in *C. sinensis* (*p* = 0.96), although, it significantly reduced the number of root tips in *C. grandis* (*p* = 0.046; Figure 1).

### 2.2. miRNA Sequencing Data Processing, Annotation, and Primary Analysis

A total of 12 cDNA libraries from the control and B toxicity-treated *C. sinensis* and *C. grandis* roots were sequenced. Clean reads ranging from 11,708,012 to 20,753,601 were retained from each library. The sRNAs in each library were annotated in Table 1.

Most of the unique clean reads in the 12 sRNAs libraries were between 21 and 24 nt. The read counts with 24 nt were the greatest, followed by those with 21 nt. Analysis revealed that 12.25% of the 24 nt sequences, yet only 7.30% of the 21 nt sequences, in the 12 sequenced libraries mapped to the *C. clementina* genome (Appendix A).

For known miRNA identification and novel miRNA prediction, we used BLAST searches and the MTide program, respectively, and a total of 172 mature miRNAs were identified from all the clean reads of the 12 sRNA libraries. Among the identified miRNAs, five were annotated *Citrus* miRNAs (in the miRbase v21.0), while 167 were novel miRNAs (133 conserved and 34 non-conserved; Appendix A). As reported in B toxicity-treated *Citrus* leaves [19], single nucleotide polymorphisms (SNP) were detected in the mature miRNA sequences.

Thirty-one of the novel non-conserved miRNAs could be considered new miRNAs since they had a definite anti-sense miRNA (miRNA*) sequence. The remaining three non-conserved miRNAs might still be considered potential novel miRNAs in *Citrus* based on their dominant read counts of above 700, even though none of corresponding miRNA* sequence was found in our dataset (Appendix A).

Intriguingly, of the novel mature miRNAs identified in this study, 71 (41.28%) were of 24 nt in length, while 59 (34.30%) were of 21 nt (Appendix A), indicating that 24 nt-long miRNAs are predominant in *Citrus* roots, which was in line with previous reports by Lu et al. [17] on B deficiency-treated *C. sinensis* roots, although it was different from the results of B toxicity-treated *Citrus* leaves [19]. As seen in Appendix A, 69.64 % of the 21-nt-long novel miRNAs began with a 5′ uridine, whereas 66.19 % of those with 24 nt started with a 5′ adenosine. This was consistent with our previous report for B toxicity-treated *Citrus* leaves [19].

### 2.3. Differential Expressions of miRNAs in Response to B Toxicity

The expression patterns of miRNAs were slightly different between *C. sinensis* and *C. grandis*. Specifically, miR7496 and another three non-conserved miRNAs (‘scaffold_3_10325′, ‘scaffold_8_42593′, and ‘scaffold_8_42594′) were almost undetectable in *C. grandis* roots; miR3951 and a non-conserved ci-miRN17 were expressed consistently under control and B-toxic treatments at high levels in *C. sinensis* roots, and miR9563 in *C. grandis* roots (Appendix A).

Further analysis revealed that 37 (23 up- and 14 down-regulated) and 11 (six up- and five down-regulated) miRNAs were differentially expressed in response to B toxicity in *C. grandis* and *C. sinensis*, respectively (Appendix A). It is worth mentioning that the expression patterns of most of the B-toxicity-responsive miRNAs were similar between *C. sinensis* and *C. grandis*, and that among the above mentioned seven miRNAs that expressed in a species-specific manner, miR3951, miR7496, and ci-miRN17 were significantly down-regulated in *C. grandis* roots.

### 2.4. Prediction and GO Analysis of Targets for DE miRNAs

Using the TargetFinder, a total of 101 genes were predicted for the 172 identified conserved and non-conserved miRNAs. Given the specificity of miRNA expression in plants and animals [20,21]; Pobezinsky et al. [22] specifically expressed that miRNAs in either *C. sinensis* or *C. grandis* roots might be involved in the different response to B toxicity between the two *Citrus* species. However, only three targets were obtained for the seven species-specific miRNAs. Among them, *Ciclev10028368m.g.v1.0* was targeted by miR3951, whereas both *Ciclev10029732m.g.v1.0* and *Ciclev10024645m.g.v1.0* were co-targeted by ‘scaffold_8_42593′ and ‘scaffold_8_42594′. Unfortunately, these targets were poorly annotated; we therefore focused on candidate targets predicted from miRNAs that showed a discrepancy between control and B-toxic treatments in each species. Finally, 29 target genes were predicted from the 47 (37 for *C. grandis*, 11 for *C. sinensis*, and one shared) B-toxicity-responsive miRNAs (Appendix A).

GO annotation of the differentially expressed (DE) miRNA targets were applied for enrichment analysis. However, no significant shared GO term was obtained because all the expected gene counts were less than five. Nevertheless, it is noteworthy that most of candidate targets were enriched in categories classified as biological process, and that these targets were predicted from miRNAs that were only differentially expressed in *C. grandis* roots, indicating that B-toxic treatment has obviously greater impact on *C. grandis* roots compared to *C. sinensis* (Appendix A).

### 2.5. qRT-PCR Relative Expression Analysis of DE miRNAs and Target Genes

For validation of the Illumina sequencing results, relative expression levels of 22 randomly selected miRNAs were determined by stem-loop qRT-PCR. However, the qRT-PCR programs of three selected miRNAs were aborted due to nonspecific amplification of their corresponding primer pairs. Among the other 19 (17 conserved and two non-conserved) miRNAs, miR156, and miR159 were both down-regulated in *C. grandis* and *C. sinensis* under B toxicity when assayed via qRT-PCR, whereas they were all up-regulated according to direct sequencing; miR395a and miR167-3p were down-regulated in response to B toxicity in *C. grandis*, although they were up-regulated according to Illumina sequencing. Overall, in the two *Citrus* species, 71.05% of the successfully tested miRNAs had a similar expression pattern between the qRT-PCR and the direct sequencing results in response to B toxicity (Figure 2; Appendix A).

Sixteen target genes predicted from eight DE miRNAs were assayed via qRT-PCR. Six candidate genes co-targeted by miR156 and miR156k, including *Ciclev10031270m.g.v1.0*, *Ciclev10031391m.g.v1.0*, *Ciclev10020532m.g.v1.0*, *Ciclev10031834m.g.v1.0*, *Ciclev10011938m.g.v1.0*, and *Ciclev10004527m.g.v1.0* showed the expected changes in mRNA levels in *C. sinensis* roots, yet their expression levels were not significantly changed in *C. grandis* roots. The levels of *Ciclev10009458m.g.v1.0* and *Ciclev10000756m.g.v1.0* were negatively correlated with those of miR398 and miR319, respectively, in both *C. sinensis* and *C. grandis* roots, indicating that they might be regulated via miRNA-mediated cleavage. The expression of *Ciclev10019529m.g.v1.0* was positively correlated with the level of miR396g-5p. The remaining target genes maintained a relatively stable expression level (Table 2; Figure 2 and Figure 3). Overall, our results, as previously reported by Lu et al. [17] for *Citrus* and Moldovan et al. [23] for *Arabidopsis*, showed that profiling target gene expressions by qRT-PCR is not an efficiency way to predict miRNA targets.

### 2.6. Experimental Validation of miRNA-Mediated Cleavage of Target Genes

Seventeen candidate targets (including all the 15 annotated ones) were experimentally verified via modified 5′-RACE analysis. Finally, *Ciclev10000756m.g.v1.0*, *Ciclev10018916m.g.v1.0*, and *Ciclev10003452m.g.v1.0*, which encode a myeloblastosis (MYB) transcription factor, a SCARECROW-like protein, and a cation transporting ATPase, respectively, were confirmed to be cleaved by their corresponding miRNAs (Figure 3). Intriguingly, *Ciclev10000756m.g.v1.0*, which was co-targeted by miR319 and miR159, produced only one product, while *Ciclev10018916m.g.v1.0*, which was simultaneously targeted by miR171, miR171b, miR171f, and ctr-miR171, generated three products with different abundances.

### 2.7. Bioinformatics of Ciclev10000756m.g.v1.0 Encoded Protein

The biological function of *Ciclev10000756m.g.v1.0* in *Citrus* roots is still not understood. Given the various functions of *MYBs′* family members in plant growth and development, bioinformatic analysis of *Ciclev10000756m.g.v1.0* may help us to further understand its role in *Citrus* response to B toxicity. The phylogenetic relationships within *Ciclev10000756m.g.v1.0* and 43 randomly selected *AtMYBs*, based on multiple amino acid sequence alignment, clustered them into five different groups, in one of which *Ciclev10000756m.g.v1.0*, *AtMYB96, 44*, *77*, *61*, and *93* were present (Figure 4). Interestingly, *AtMYB96, 44*, *77*, *61*, and *93* were all involved in the regulation of lateral root growth and development in *Arabidopsis* [24,25,26,27,28], implying that *Ciclev10000756m.g.v1.0* might also function in *Citrus* lateral root development.

### 2.8. Non-Structural Carbohydrates in Citrus Roots

Two-way between-groups ANOVA was used to examine the main effects and interactions of B treatment and species for starch, fructose, glucose, and sucrose contents. As shown in Figure 5, there were significant two-way interactions of B treatment and species for the above mentioned non-structural carbohydrates in citrus root tips (*F* = 55.140, *p* < 0.001; *F* = 29.238, *p* < 0.001; *F* = 10.623, *p* = 0.007; *F* = 12.766, *p* = 0.004; respectively). B-toxic treatment significantly increased starch, glucose, and sucrose contents in *C. sinensis* root tips and decreased starch and fructose contents in *C. grandis* roots; although, it did not impact glucose and sucrose contents in *C. grandis* roots.

## 3. Discussion

In *Citrus*, easily recognized symptoms of B toxicity only occur in leaves, but do not appear to develop in roots [5,7,8,12,29]. B-toxic treatment does not change root meristem structure in either tolerant or intolerant *Citrus* cultivars. These results were presumed to be caused by the relative lower B levels in the roots than in the leaves [8]. Similarly, our results indicated that B-toxic treatment had little impact on radial (average diameter) and axial growth (total root length) of the root system. Nevertheless, B-toxic treatment significantly changed, in *C. grandis*, the root system architecture by reducing the number of root tips (Figure 1), suggesting that B-toxicity treatment restricts lateral root differentiation in intolerant *C. grandis*. Since B-toxic treatment significantly alters the expression levels of genes [30] and abundances of proteins [31] in *Citrus* roots, the roots of intolerant species might perceive B toxicity to the aboveground parts via certain signal transduction pathways, and respond appropriately for plant adaptation.

Using Illumina sequencing, a total of 172 (five known, 133 conserved, and 34 novel) miRNAs were identified from B-toxic treated *Citrus* roots in the present study. Among the identified miRNAs, 37 (33 conserved and four novel) in *C. grandis* roots and 11 (nine conserved and two novel) in *C. sinensis* showed differential expressions after the B-toxic treatment, suggesting that miRNAs were involved in the adaptive response of root tips to long-term B toxicity.

### 3.1. Tissue-Specific Expression of miRNAs in Response to B Toxicity

Previous studies have shown tissue specificity of plant miRNA expression under various environmental stresses. For instance, *miR397* and *miR398* were specifically expressed in maize leaves and *miR319* and *miR395* in the nitrate-limited roots [32]. Similarly in barley, *miR397* were inhibited in the leaves under B toxicity, whereas *miR319* and *miR395* were induced in the roots exposed to B toxicity [18]. In B toxicity-treated *Citrus*, miR397, and miR398 were specifically expressed in the leaves [19] and roots (Figure 2), respectively; the expression of miR319 was enhanced in *C. grandis* roots (Appendix A) but was repressed in *C. sinensis* leaves [19]; miR395 was expressed in both the leaves and roots of *C. grandis* under B toxicity, but with an opposite expression pattern [19] (Figure 2). These results suggest that the tissue specificity of miRNA expression varies with plant species as well as the environmental stimuli.

The specificity of B-toxicity-responsive miR397 in *C. grandis* and *C. sinensis* leaves might be due to its specific expression in the leaves among *Citrus*. This is supported by our Illumina sequencing results of the 12 B-treated root sRNA libraries, from which not even one read count of miR397 mature sequence was obtained.

Distinctions of the expression of miR395 among plant species, as well as those between the leaves and roots in *C. grandis*, might result from its targets functions. In both maize and barley, miR395 targets genes belonging to the ATP sulfurylase gene family (also known as *APS* genes) [32,33], which modulate sulphur assimilation [34]. Since N deficiency reduces sulfate-use efficiency [35,36], and since synergism and/or antagonism exist among nutrient elements in various plants, the root-specific expression of miR395 in maize and barley under nutrient stress such as N deprivation might silence the *APS* genes, thus enhancing sulfate accumulation. Unlike this, the only candidate target of *Citrus* miR395 encodes a 3-ketoacyl-CoA thiolase 2 (KAT2), involved in the timely onset of leave senescence [37]. Given that senescence and abscission of leaves can remove excess B from the plants, the opposite expression pattern of miR395 in *C. grandis* leaves and roots might be attributed to different cell fates of the two organs, which accumulate vastly different B levels [8].

### 3.2. Validation of miRNA Targets

Since miRNAs regulate specific targets through cleavage in plants [15], RLM-5′-RACE analysis is currently one of the convincing approaches for validation of miRNA targets. Using this approach, *Ciclev10000756m.g.v1.0* (*CiMYB*), *Ciclev10018916m.g.v1.0*, and *Ciclev10003452m.g.v1.0*, were confirmed to be cleaved by the corresponding miRNAs (Figure 3).

Our results lead to two sets of questions. The first one relates to determination of the bona fide target of miRNA. The 5′-RACE approach currently cannot determine whether miR319 or miR159, or even both target *Ciclev10000756m.g.v1.0*, since miR319 and miR159 had nearly identical complementary sequence to the target. In *Arabidopsis*, miR319 and miR159 are two closely related miRNA families that share high sequence similarity, and generally, the same targets belonging to *TCPs* and/or *MYBs* family could be obtained via bioinformatics. Nonetheless, *Arabidopsis* miR319 and miR159 are found to be functionally specific for the two different sets of targets. Particularly, miR159 modulates floral development via targeting to *GAMYB* [38], whereas miR319 controls leaf morphogenesis through guiding messenger RNA cleavage of *TCPs* [39,40]. Such specificity of miRNA regulation is attributed to the expression levels and sequence differences of corresponding miRNAs [41]. Similarly, *Ciclev10000756m.g.v1.0* might be targeted by miR159. However, they did not show an expected negative change in RNA levels under the B treatments. The expression level of *Ciclev10000756m.g.v1.0*, in turn, was negatively correlated with that of miR319. The read counts of miR159 are far less than those of miR319 from the Illumina sequencing results. These together suggest that *Ciclev10000756m.g.v1.0* (*MYB*) is more likely to be regulated by miR319 in *Citrus* roots. In fact, miR319 can target both *TCPs* and *MYBs* [41]. A recent report by Zhang et al. [42] also indicated that miR319b, rather than miR159, targets *MYB33* for degradation specifically in *Arabidopsis* roots exposed to ethylene, leading to root growth inhibition.

The second question concerns the post transcriptional regulation of miRNA target. It is widely accepted that plant miRNAs bind to target transcripts for mRNA decay or translational repression. MiRNA-mediated decay of target mRNA generally results in a negative relationship of transcription level between the miRNAs and their targets [34]. However, some miRNAs still show a positive temporal correlation with their target genes [34,43]. Such contradiction is supposed to be resulted from their different spatial expression [34], which restricts miRNA-mediated cleavage of the target mRNAs. Even though the RNA level of *Ciclev10018916m.g.v1.0* was not correlated with that of any of the miR171 family member, the former was definitely cleaved, guided by different members of the later, given that three tagged mRNA fragments, 5′-ends of which would precisely correspond to the tenth nucleotide of different miRNA complementary sites [44], were detected via the 5′-RACE approach (Figure 3B). As a result, significant down and/or up regulation of the miR171 family members in response to B toxicity (Appendix A) allowed the maintenance of *Ciclev10018916m.g.v1.0* expression at a stable level in *C. grandis* roots under B toxicity.

### 3.3. B-Toxicity-Responsive miRNAs in Citrus Roots

The numbers of B-toxicity-responsive miRNAs for *C. grandis* and *C. sinensis* roots (37 and 11, respectively) were less than those for leaves (51 and 20, respectively; [19], suggesting, again, a relatively mild B-toxic effect in the roots.

Among the DE miRNAs, miR319 were significantly up-regulated in *C. grandis* roots when exposed to B toxicity (Figure 2B), repressing the expression of *Ciclev10000756m.g.v1.0* (Figure 3B), which is homologous to *AtMYB96, 44*, *77*, *61*, and *93* (Figure 4). In *Arabidopsis*, MYB96 has been proposed to be a molecular link that mediates ABA–auxin cross talk in lateral root growth under drought stress [24]; MYB44 and MYB77 interact directly with auxin-responsive regulator genes, controlling lateral root formation under changing environmental conditions [25,26], while MYB93, which is also required for normal auxin responses, interacts with ARABIDILLO proteins in the endodermal cells overlying early lateral root primordial, negatively regulating lateral root development [28]. Unlike these, *At*MYB61 appears to allocate carbon to non-recoverable sinks, but meanwhile promotes lateral root growth under favorable conditions [27]. Therefore, it is reasonable to assume that *Ciclev10000756m.g.v1.0* might also function in the regulation of lateral root development in *Citrus*. The down-regulation of *Ciclev10000756m.g.v1.0* due to up-regulation of miR319 in B toxicity-treated *C. grandis* roots might decrease lateral root formation, thus dramatically changing the root system architecture (Figure 1).

Using 5′-RACE, we confirmed *Ciclev10018916m.g.v1.0* as a real target of miR171 family members. SCARECROW is the first GRAS transcription factor discovered in plants [45]. It delimits SHORTROOT movement in the endodermis via protein–protein interaction [46], thus regulating an asymmetric endodermal cell division that is essential for generating the radial organization of the plant root [45]. The maintenance of *Ciclev10018916m.g.v1.0* expression resulting from up- and/or down-regulation of miR171 family members in B toxicity-treated *Citrus* roots might fulfil SCARECROW/SHORTROOT interaction, which plays an important role in stem cell maintenance, and quiescent center and endodermis specification [46]. Indeed, long-term B-toxic treatment seems not to affect the tissue organization in *Citrus* root tips [8], allowing regular root elongation under B toxicity (Figure 1).

### 3.4. Adaptive Responses of Citrus Roots to Long-Term B Toxicity

Water and nutrient acquisition in plants relies largely upon the root system architecture [47]. Reduction of lateral root number in tetraploid Carrizo citrange (*C. sinensis* × *Poncirus trifoliata*) plants can significantly decrease the absorption and transport of B, lowering B concentration in the aboveground parts under B toxicity [48]. However, expected mitigation for B toxicity in B toxicity-treated *C. grandis* leaves was not observed [8], even though the root system architecture was dramatically changed (Figure 1). Given that the activity of *AtMYB61*, which is homologous to *Ciclev10000756m.g.v1.0* targeted by miR319, is clearly modulated by photoassimilates such as sugar [49,50], one hypothesis is that the decrease of lateral root formation modulated by miR319 in intolerant *C. grandis* might be a response to carbohydrate starvation in the roots, rather than directly to the B treatment, because long-term B toxicity specifically triggers programmed cell death (PCD) of leaf phloem tissue [8] that could impairs photoassimilate translocation to roots in *C. grandis* (Figure 5). The reduction of root tip numbers in *C. grandis*, in the longer run, might limit B absorption and transport. As expected, a relatively lower B concentration has been observed in long-term B toxicity-treated leaves of intolerant *C. grandis*, compared to those of tolerant *C. sinensis* [8,13,14].

## 4. Materials and Methods

### 4.1. Plant Culture and B Treatments

Eleven-week-old *Citrus sinensis* (B-tolerant) and *C. grandis* (B-intolerant) seedlings were sand-cultured and treated with Hoagland’s nutrient solution supplemented with 10 (control) or 400 (B-toxic) μM H_3_BO_3_ as described by [8]. After the B treatments for 15 weeks, root tips (approximately 5 mm) were dissected and immediately frozen in liquid N_2_ and stored at −80 °C until RNA extraction. Root tips dissected from five plants of the same B treatment were mixed as one biological replicate.

### 4.2. Determination of Root Traits

Root samples were thoroughly washed, rearranged to reduce overlap and crossing, and then scanned at 200 dpi in 20 × 30 cm trays. Root images were analyzed using the WinRHIZO software (Version 2009b, Regent, Montreal, CA, USA) in Lagarde’s mode [51].

### 4.3. RNA Preparation

Total RNA of root-tip samples was extracted with the TRIzol reagent (Invitrogen, Carlsbad, CA, USA). The total RNA from each replicate was divided into three sections, which were further used for Illumina sequencing, qRT-PCR assay, and modified 5′-RACE amplification, respectively. All the RNA samples were stored at −80 °C before use.

### 4.4. Library Construction, Illumina Sequencing, and Data Processing

About 1 µg of root total RNA from each biological replicate (three biological replicates for each treatment) was used for small RNA library construction using the TruSeq Small RNA Sample Prep Kit (Illumina, San Diego, CA, USA). Qualified libraries were then sequenced as previously described by Huang et al. [19] with an Illumina HiSeq 2500. Raw data from high-throughput sequencing were pre-processed by the Illumina pipeline filter as reported [19]. Known miRNAs were then identified via subjection of the clean reads to BLAST searches against the miRBase 21 (http://www.mirbase.org/); whereas novel miRNAs via mapping the remaining non-annotated sequences to the *C. clementina* genome, (http://www.phytozome.org/clementine.php) as previously reported [19].

### 4.5. Analysis of Differentially Expressed (DE) miRNAs

Both the *p*-value and fold-change of each miRNA identified from the 12 libraries were calculated using DESeq software [52]. The false discovery rate (FDR) were then generated from the *P*-value based on the Benjamini Hochberg Method. A 1.5-fold cut-off together with an FDR of less than 0.01 was applied for determination of DE miRNAs.

### 4.6. Prediction and GO Analysis of miRNA Target Genes

Target genes of miRNAs obtained above were predicted using TargetFinder according to Allen et al. [53] and Schwab et al. [39]. GO terms of all predicted targets were then analyzed with the GO database (http://www.geneontology.org/).

### 4.7. QRT-PCR of DE miRNAs and Candidate Targets

For verification of DE miRNAs obtained from sequencing, 22 miRNAs were randomly selected to perform stem-loop qRT-PCR using *U6* snRNA as an internal control [54]. To monitor the relative abundance of candidate targets, qRT-PCR reactions were performed on a Mastercycler Ep Realplex System (Eppendorf, Hamburg, Germany) using *actin* (AEK97331.1) as an internal control. Primers used for stem-loop qRT-PCR, as well as those for qRT-PCR of target genes were designed as reported by Huang et al. [19] and listed in Appendix A, respectively. More details for RT-PCR can be found in Appendix A. Tests were performed with three biological replicates with two technical repeats. Differences between B treatments were analyzed by IBM SPSS software using independent sample *t*-test.

### 4.8. Validation of miRNA Targets via RNA Ligase-Mediated 5′-RACE

Poly (A)^+^ mRNA was isolated from root total RNA with the polyAtract mRNA isolation kit (Promega, Madison, WI, USA), followed by RNA ligase-mediated 5′-RACE with the GeneRacer Kit (Invitrogen, Carlsbad, CA, USA) as described by Song et al. [55]. The gene-specific and nested primers were designed according to Huang et al. [19] and were listed in Appendix A. The nested PCR products were finally recovered from agarose gel, cloned into T-vectors, and sequenced.

### 4.9. Sequence Analysis and Phylogeny of MYBs

The *MYB* homologues targeted by miR319 and/or miR159 were identified using BLASTP from *C. clementina* genome, while 43 randomly selected *MYB* family members of model plant were identified using BLASTP from fully sequenced *Arabidopsis thaliana* genomes via GenBank. Sequences were aligned using MEGA (Version 7.0.21). The phylogenetic tree was calculated using the maximum likelihood algorithm.

### 4.10. Measurements of Sugar Contents

Starch, glucose, fructose, and sucrose contents in root tips treated with different B levels were extracted three times with 80% (*v*/*v*) ethanol at 80 °C (3 mL each, 30 min per extraction), and determined as previously described by Han et al. [56]. Results of five biological replicates were analyzed by IBM SPSS software using univariate two-way ANOVA, followed by Sidak Post-Hoc Test.

## 5. Conclusions

Here, using Illumina sequencing, 37 (23 up- and 14 down-regulated) and 11 (six up- and five down-regulated) DE miRNAs were identified from B toxicity-treated *C. grandis* and *C. sinensis* roots, respectively. Among them, miR319, miR171, and miR396g-5p were confirmed by the qRT-PCR and 5′-RACE approaches to target a MYB transcription factor gene, a SCARECROW-like protein gene, and a cation transporting ATPase gene, respectively. Through integrating the current finding with previous ones, we put forward a miRNA-mediated adaptive mechanism to long-term B toxicity in *Citrus* roots. In summary, the miR171 family members modulated SCARECROW expression that functions in stem cell maintenance, quiescent center, and endodermis specification, allowing regular root elongation under B toxicity, whereas miR319, in intolerant *C. grandis*, might respond to carbohydrate starvation generated by B toxicity to the aboveground parts, reducing MYB expression and suppressing lateral root formation, thereby limiting B absorption and upward transport, which may improve plant growth under long-term B toxicity.

## Figures and Tables

**Figure 1 ijms-20-01422-f001:**
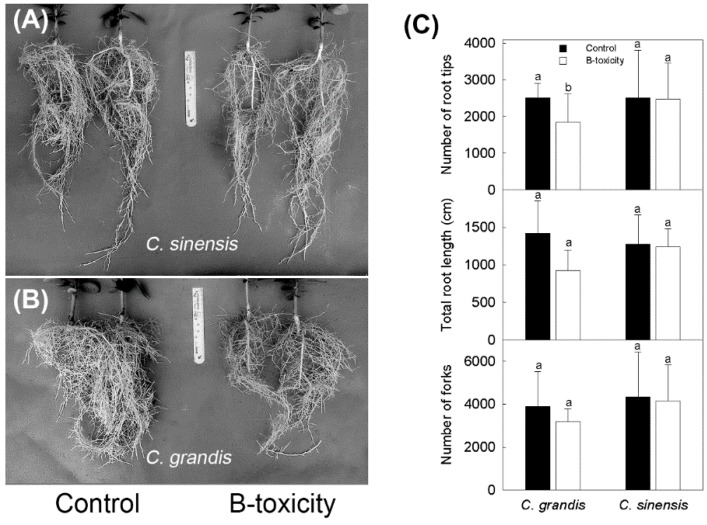
B toxicity affected Citrus root architecture. (**A**) Boron (B)-treated *C. sinensis* roots, (**B**) B-treated *C. grandis* roots, (**C**) determination of root traits. Results represent mean ± SD (*n* = 8 and 5 for *C. grandis* and *C. sinensis*, respectively). Different letters above columns indicate significant difference at *p* < 0.05 according to independent sample *t*-test.

**Figure 2 ijms-20-01422-f002:**
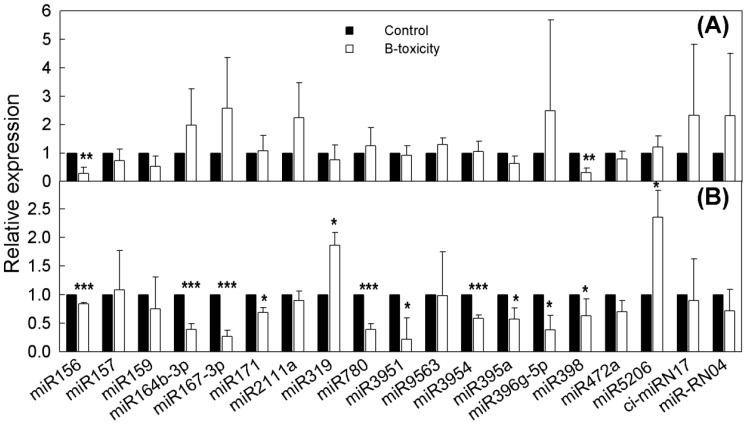
Relative abundance of microRNAs (miRNAs) between control and B toxicity in *C. sinensis* (**A**) and *C. grandis* (**B**) root tips. microRNA levels in control and B-toxic root tips were determined using stem-loop qRT-PCR. Results represent mean ± SD (*n* = 3). “*”, “**”, and “***” indicate significant difference at *p* < 0.05, 0.01, and 0.001 level, respectively, according to independent sample *t*-test.

**Figure 3 ijms-20-01422-f003:**
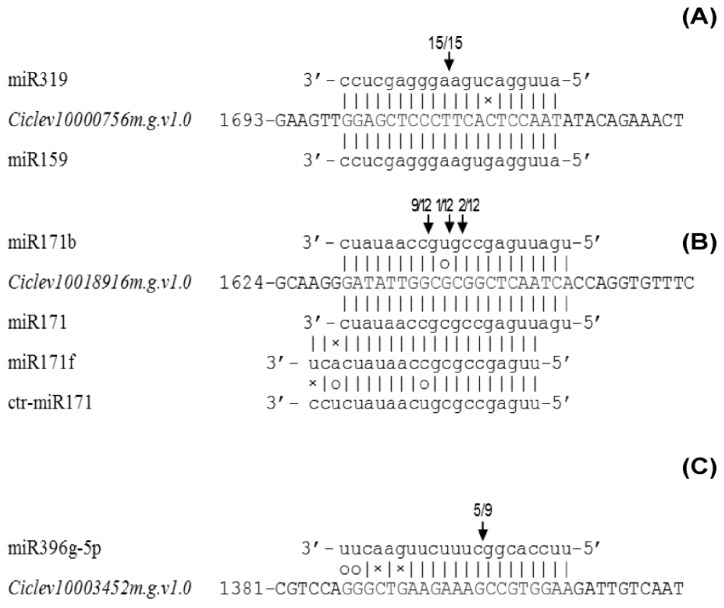
qRT-PCR expression profiles and modified 5′-RACE validation of the mRNA targets. (**A**), *Ciclev10000756m.g.v1.0*, (**B**), *Ciclev10018916m.g.v1.0*, and (**C**), *Ciclev10003452m.g.v1.0*. Cleavage sites (vertical arrows) of targeted mRNA were determined by RLM-5′-RACE, and the frequencies of clones were shown.

**Figure 4 ijms-20-01422-f004:**
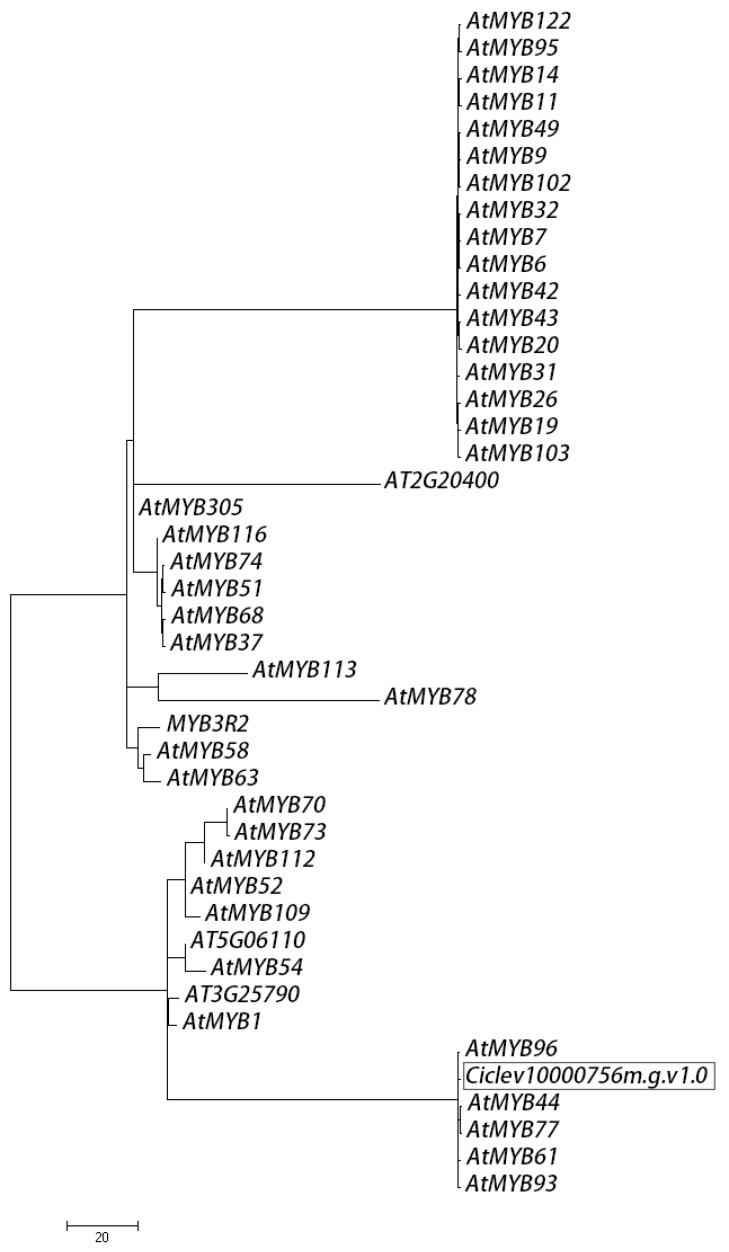
Inferred phylogenetic tree (maximum-likelihood) of full-length *AtMYB* proteins identified in *A. thaliana* and *Ciclev10000756m.g.v1.0*. Protein IDs used in the initial sequence alignment were obtained from the GenBank and the *clementina* genome, respectively.

**Figure 5 ijms-20-01422-f005:**
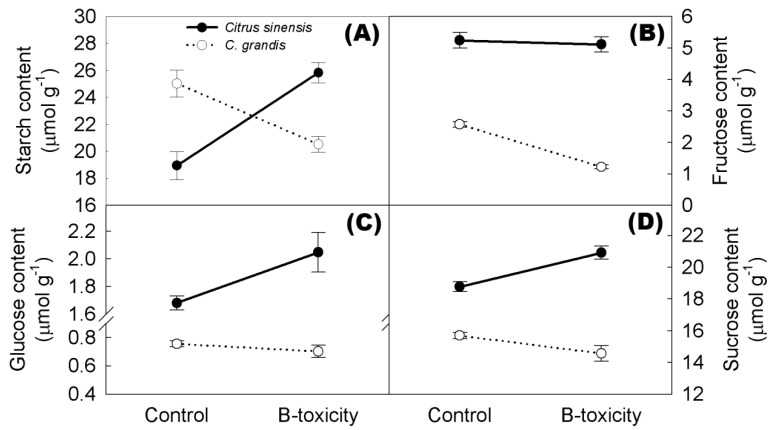
Effects of B toxicity on starch (**A**), fructose (**B**), glucose (**C**), and sucrose (**D**) contents in citrus roots. Results represent means ± SE (*n* = 6).

**Table 1 ijms-20-01422-t001:** Annotation of sRNAs from *Citrus* root tips treated with sufficient and toxic level of B.

	*C. grandis*						*C. sinensis*					
	B-Toxic Treated			Control			B-Toxic Treated			Control		
	Replicate 1	Replicate 2	Replicate 3	Replicate 1	Replicate 2	Replicate 3	Replicate 1	Replicate 2	Replicate 3	Replicate 1	Replicate 2	Replicate 3
Clean read counts	17,034,957	13,517,774	12,734,477	20,753,601	17,185,770	16,082,626	12,685,529	14,237,047	15,683,586	11,708,012	13,599,502	13,102,318
rRNA	41.27%	47.68%	44.15%	43.45%	34.82%	44.06%	46.80%	42.51%	45.47%	50.73%	42.24%	53.03%
scRNA	0.00%	0.00%	0.00%	0.00%	0.00%	0.00%	0.00%	0.00%	0.00%	0.00%	0.00%	0.00%
snRNA	0.00%	0.00%	0.00%	0.00%	0.00%	0.00%	0.00%	0.00%	0.00%	0.00%	0.00%	0.00%
snoRNA	0.18%	0.16%	0.17%	0.11%	0.17%	0.18%	0.14%	0.17%	0.20%	0.14%	0.17%	0.13%
tRNA	2.88%	2.65%	3.26%	2.84%	2.92%	2.45%	2.74%	2.70%	3.34%	3.00%	3.10%	2.93%
Repbase	0.07%	0.07%	0.09%	0.06%	0.07%	0.08%	0.08%	0.08%	0.12%	0.08%	0.08%	0.08%
Unannotated	55.60%	49.44%	52.33%	53.54%	62.02%	53.23%	50.24%	54.53%	50.87%	46.05%	54.42%	43.84%

**Table 2 ijms-20-01422-t002:** Relative expressions of predicted miRNA targets in B-toxic treated *Citrus* root tips by qRT-PCR.

miRNAs	Target ID	Annotation	Relative Expression (Mean ± SD)	
			*C. sinensis*	*C. grandis*
miR156	Ciclev10032171m.g.v1.0	GTPase domain	1.15 ± 0.55	1.39 ± 1.23
	Ciclev10031270m.g.v1.0	SBP domain	0.80 ± 0.19 *	0.91 ± 0.18
	Ciclev10031391m.g.v1.0	SBP domain	0.58 ± 0.39 *	0.72 ± 0.59
	Ciclev10020532m.g.v1.0	SBP domain	0.54 ± 0.43 *	0.96 ± 1.26
	Ciclev10031834m.g.v1.0	SBP domain	0.57 ± 0.38 *	1.54 ± 1.40
	Ciclev10011938m.g.v1.0	SBP domain	0.47 ± 0.41 *	1.05 ± 0.45
	Ciclev10004527m.g.v1.0	Protein kinase domain	0.88 ± 0.40 *	0.84 ± 0.46
miR171	Ciclev10019083m.g.v1.0	SCARECROW-like protein	1.09 ± 0.32	1.44 ± 0.50
	Ciclev10019094m.g.v1.0	SCARECROW-like protein	1.27 ± 0.24 *	1.25 ± 0.42
	Ciclev10018916m.g.v1.0	SCARECROW-like protein	1.36 ± 0.78	1.44 ± 0.87
miR319	Ciclev10014986m.g.v1.0	TCP (TEOSINTE BRANCHED1/CYCLOIDEA/PROLIFERATING CELL FACTOR) transcription factor	1.57 ± 0.95	0.97 ± 0.70
miR319/miR159	Ciclev10000756m.g.v1.0	MYB transcription factor	1.32 ± 0.86	0.40 ± 0.22 *
miR3951	Ciclev10028368m.g.v1.0	Leucine rich repeats	0.89 ± 0.50	0.90 ± 0.65
miR396g-5p	Ciclev10019529m.g.v1.0	WRC (Trp–Arg–Cys) growth-regulating factor 3	1.29 ± 0.91	2.17 ± 1.42
	Ciclev10003452m.g.v1.0	Cation transporting ATPase	1.32 ± 0.95	0.33 ± 0.26 *
miR398	Ciclev10009458m.g.v1.0	Plastocyanin-like domain	3.33 ± 2.35 *	1.52 ± 0.98

Note: All the values were expressed relative to the control. “*” indicates a significant difference at *p* < 0.05 level, according to independent sample *t*-test. MYB is myeloblastosis.

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
