# Peer review of "MicroRNA Sequencing Revealed Citrus Adaptation to Long-Term Boron Toxicity through Modulation of Root Development by miR319 and miR171"

_ijms, 2019, doi:10.3390/ijms20061422_

Reviewer 1 Report

In this manuscript, the authors investigated the involvement of miRNA in response of Citrus root to Boron toxicity. They first show that the effect of B treatment on root system architecture is limited to a reduction of root tip in the sensible specie of Citrus (C. grandis) and confirm the absence of  effect on the resistant one (C. sinensis). The authors sequence small RNA from root grown in control or Boron treated of these two species of Citrus. They identified 172 miRNAs, the majority being newly described in Citrus. Most of differentialy accumulated miRNA in response to B treatment were the same between the two different Citrus species. The differential level of expression of miRNAs was confirmed by stemloop qRT-PCR. The authors predicted the targets of the identified miRNAs and confirmed expected expression anti-correlation for 3 differentially expressed miRNAs. Cleavage of the targets was confirmed for 3 genes targeted by 3 family of miRNAs.

Remarks

line 105-108: it would have been nice to have distribution of unique reads and of all reads to give feedback on the diversity (unique) compared to the abundance (all reads).It is not clear for me which read were considered for the prediction of miRNA: unique reads (said in the beginning of the paragraph), uniquely mapped reads or all reads mapping on unannotated regions of the genome.
- considering only unique sequence do not make sense since miRNA are characterized by the high accumulation of a given sequence
- if limiting to uniquely mapped reads, it will discard a great number of miRNA families that give rise to the same mature miRNA from different loci.
- The last choice make the more sense for me.
A clarification of the text is required.

GO analysis need to be better described. What are the p-value of enrichment for each displayed GO?

Minor remarks

Instructions for the writing of the manuscript are still included for the introduction.

line 180: reference should be Table 2; Figure 2 and 3.

Table 2: I would add the value of the genes in Figure 3 to Table 2 for an easier interpretation of the results.

line 232: radial growth is average diameter and axial growth is total root length

line 252: buy -> but

line 378: certainly far lower than 1 for FDR cutoff: 0.01?

Author Response

Point 1: line 105-108: it would have been nice to have distribution of unique reads and of all reads to give feedback on the diversity (unique) compared to the abundance (all reads).It is not clear for me which read were considered for the prediction of miRNA: unique reads (said in the beginning of the paragraph), uniquely mapped reads or all reads mapping on unannotated regions of the genome.

- considering only unique sequence do not make sense since miRNA are characterized by the high accumulation of a given sequence

- if limiting to uniquely mapped reads, it will discard a great number of miRNA families that give rise to the same mature miRNA from different loci.

- The last choice make the more sense for me.

A clarification of the text is required.

Response 1: Indeed, we used all the clean reads (rather than unique clean reads) for miRNA identification. To identify known miRNAs, the clean reads from the 12 sRNA libraries were aligned with known miRNAs from other plant species in miRBase 21. The remaining un-annotated sequences (mapping on un-annotated regions of the clementine genome) were used for novel miRNA prediction (These information could be found in line 377-380). line 105-108 were only description of the primary analysis result of the sequencing data. To avoid puzzling readers, paragraphs related to miRNA identification have been rewrite in the revised MS.

Point 2: GO analysis need to be better described. What are the p-value of enrichment for each displayed GO?

Response 2: We obtained only a small number of differentially expressed miRNA targets in this study, some of which have no definit GO annotation. When uploading our GO data of predicted targets into the web server, the p-value of enrichment for each displayed GO were indicated as “MI”, which stands for “meaningless” because that the expected gene counts for each GO term enriched are all less than 5. Therefore, we could not have any further description of the GO analysis results. We simply compare the impact of B-toxicity between the two Citrus species by comparing the number of targets enriched. To make it clearly, Line 152-157 has been rewritten.

Minor remarks

Instructions for the writing of the manuscript are still included for the introduction.

Response: Relevant paragraph have been deleted accordingly.

line 180: reference should be Table 2; Figure 2 and 3.

Response: We’ve confirmed and corrected corresponding sentence as mentioned by the reviewer.

Table 2: I would add the value of the genes in Figure 3 to Table 2 for an easier interpretation of the results.

Response: The expression levels of three genes targeted by miR171 have been added to Table 2. And their expressions were deleted in revised Figure 3.

line 232: radial growth is average diameter and axial growth is total root length

Response: Accepted.

line 252: buy -> but

Response: Accepted.

line 378: certainly far lower than 1 for FDR cutoff: 0.01?

Response: Accepted.

Reviewer 2 Report

The paper entitled "MicroRNA sequencing revealed Citrus adaptation to long-term boron toxicity through modulation of root development by miR319 and miR171" by Huang et al., is very informative and this study is in continuation of their previous studies.

I have few questions regarding this manuscript.

1. Authors used 10 μM H3BO3 as a control treatment and 400 μM H3BO3 as a toxic concentration. In order to make a good comparison, I think the authors should take untreated plant as control. in this way the comparison will be more accurate and worthy.

2. Materials and methods lack the information about the statistical analysis.

3. Authors need to improve the language of the manuscript. sometimes the explanations in the results section are very confusing due to improper English. Authors need the help of English native speaker. 

Author Response

1. Authors used 10 μM H3BO3 as a control treatment and 400 μM H3BO3 as a toxic concentration. In order to make a good comparison, I think the authors should take untreated plant as control. in this way the comparison will be more accurate and worthy.

Response 1: Did the reviewer refer to treatment without H3BO3? Then that would be boron deficient treatment, rather than the control. Nevertheless, it would be impossible for us to obtain that data in a short time since the plant culture treatment takes one year, let alone microRNA sequencing and data analysis.

2. Materials and methods lack the information about the statistical analysis.

Response 2: Corresponding section has been added in the Materials and methods of the revised manuscript (Line 398-400, and 416-418).

3. Authors need to improve the language of the manuscript. sometimes the explanations in the results section are very confusing due to improper English. Authors need the help of English native speaker. 

Response 3: The manuscript has been checked and re-checked thoroughly. We believe that the language is now acceptable for publication.